# Quantification of Site Layout and Filter Characteristics on Primary Filter Airflow Reduction on Commercial Swine Sites in Iowa

**Benjamin Smith [1],\*, Steven Hoff [1], Jay Harmon [1], Daniel Andersen [1], Jeffrey Zimmerman [2] and John Stinn [3]** 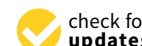

[1]   Department of Agricultural and Biosystems Engineering, Iowa State University, Ames, IA 50011, USA; hoffer@iastate.edu (S.H.); jharmon@iastate.edu (J.H.); dsa@iastate.edu (D.A.)

[2]   Department of Veterinary Diagnostic and Production Animal Medicine, Iowa State University, Ames, IA  50011, USA; jjzimm@iastate.edu

[3]   Iowa Select Farms, Iowa Falls, IA  50126, USA; jstinn@iowaselect.com

\*   Correspondence: bcsmith1@iastate.edu

**Abstract:** Fresh air intake filtration is used on commercial swine breeding-gestation-farrowing farms to reduce the frequency of airborne infectious agents. For swine producers, porcine reproductive and respiratory syndrome virus (PRRSV), influenza A virus and Mycoplasma hyopneumoniae are considered the most economically challenging airborne pathogens. Reduced frequency of disease outbreaks has been attributed to retrofitting existing systems with filtration. Economic analysis of operating costs includes energy use, maintenance and replacement of filters. Filter replacement, the largest operational cost, is dependent on filter lifespan. However, limited data is available on filter lifespan and the rate of airflow reduction during the high dust-loading periods typically encountered for filtered swine building ventilation systems. Therefore, the objectives of this study were (1) estimate the average primary filter airflow reduction per day, (2) identify the impact of factors related to site layout, filter characteristics and weather on airflow reduction rates of filters in positive-pressure ventilated buildings and (3) determine methods for reducing average primary filter airflow reduction rate per day during row-crop harvest season. Both filter brand and the installed orientation of the filter significantly ($p = 0.0314$, $p = 0.0419$, respectively) impacted airflow reduction rates. All site layout factors were significant (driveway side, $p = 0.001$; dormer orientation, $p = 0.0001$; and dormer configuration, $p = 0.0001$). The materials tested significantly reduced the airflow reduction rate during row-crop harvest. The information obtained in this study will aid producers when planning for filtration, highlight details relevant to the purchase and installation of filters, identify factors that affect filter lifespan and identify methods for improving filter lifespan.

**Keywords:** air filtration; filter lifespan; filter loading; positive pressure ventilation

## 1. Introduction

Porcine reproductive and respiratory syndrome virus (PRRSV) is an economically detrimental disease for US swine producers since first observed during the late 1980s. In 2012, it was estimated to cost the US industry USD 664 million [1,2]. Long distance aerosol transmission of PRRSV has been documented on commercial production sites, as well as the efficacy of air filtration to prevent the transmission of PRRSV via aerosols [3,4]. The use of air filtration on commercial livestock facilities was first proven effective on poultry barns to prevent the spread of Mareck's disease nearly 40 years ago [5,6]. The implementation of air filtration on commercial swine breeding-gestation-farrowing sites

has only recently become common in the US [7]. The main driving factor in the recent implementation of filtration is the economic benefit of reducing PRRSV outbreaks from aerosol transmission.

The implementation of air filtration on existing sites presents a large capital investment to the producer, with a potential payback period of 5 to 7 years [8]. The most recent economic analysis covered the capital investment for air filtration systems but only analyzed the cost of installing such a system [8]. Air filters also present an operational cost to producers, such as, added energy use, maintenance and cost of filter replacement. The addition of air filters to a ventilation system can add energy costs due to the higher operating pressure of the ventilation system, which further increases as the filters load with dust [9]. The maintenance and replacement costs of filters is highly dependent on the frequency of filter replacement and is affected by the environment and dust loads to which the filters are exposed [10]. Thus, the complete economic analysis of air filtration in swine production systems mandates the inclusion of the operational costs of filtration systems.

The largest undocumented factor for determining the operational costs of air filtration is filter lifespan but there is no published information on the lifespan of air filters on swine sites. For commercial roof-top heating, ventilation and air conditioning (HVAC) units in urban settings, data has been gathered on the dust concentrations allowing for the prediction of filter lifespan [11]. While this is an appropriate method for HVAC roof top units, swine buildings have air intakes that are near ground level that is, where dust concentration and particle size distributions are drastically different, along with different operating pressure drops and airflow rates. To address the lack of data, this study measured the airflow reduction rate of primary filters on commercial swine sites in central Iowa that utilized positive pressure ventilated buildings. The objectives of this study were: (1) determine an average primary filter airflow reduction rate per day, (2) quantify the impact of site layout, filter brand and weather on the airflow reduction rate of such filters and (3) determine methods for reducing average primary airflow reduction rate per day during row crop harvest season.

## 2. Materials and Methods

### 2.1. Site Descriptions

Data for objectives 1 and 2 were collected from eight commercial breeding-gestation-farrowing sites located in central Iowa. Each site contained six buildings, all orientated east to west, consisting of two farrowing buildings and four breeding/gestation buildings. Sites varied as to which buildings were on the north side (farrowing or breeding-gestation) and whether the driveway was on the east or west side (Figure 1). For objective 3, a commercial 3-building gilt development unit (GDU) was utilized with close proximity to row crop fields and a grain handling facility. Of the three buildings on the site, buildings 1 and 2 were utilized in the study and measured 37 m (W) by 84 m (L). Buildings 1 and 2 had air intakes on the east and west gable ends of the building. The distance between the east air intake of building one and the grain bin to the east was 29 m (Figure 2). All sites utilized a positive-pressure filtered ventilation system. Fresh air entry for each building was via two dormers extended off the side of the barn or each (two) gable end. For both designs, fresh air was pulled through an adjustable inlet curtain, evaporative cooling pad, filter bank (consisting of primary and secondary filters) and by a bank of variable speed fans. The variable speed fans exhaust pressurized the attic of the building, from which the air was distributed into the human- and animal-occupied zones through ceiling inlets. Air was exhausted from the building on the opposite side of the building from the dormer through shutters with an adjustable baffle to control the exhaust area opening. Each building had two dormers, one on the east and one on the west half of the building. The filter bank in the dormers consisted of a vertical filter wall with 6 rows of filters, thereby resulting in a floor-to-ceiling height that averaged 4.57 m (15 ft). The number of columns in each dormer varied depending on the barns capacity resulting in dormer lengths that ranged from 18.3 m to 23.2 m (60 ft to 76 ft) with a range of 144 to 216 filters. Each filter bank utilized MERV 15 v-pocket secondary filters with either a synthetic media with electrostatic charge or fiberglass media, 0.61 m by 0.61 m by 0.30 m (24 in. by 24

in. by 12 in.) with a high capacity MERV 8 primary filter with a fiberglass media, 0.61 m by 0.61 m by 0.05 m (24 in. by 24 in. by 2 in.) (Figure 3). All primary filters in the study met the same specifications previously mentioned. The primary filter brands varied only in the quality of manufacturing of the cardboard frame, pleating and gluing of the media to the cardboard. The primary and secondary filters time in operation of site ranged from six months to one week at the time of the study.

On the GDU site, the location of the test materials was randomly selected such that a test material and a control group was located on each building on alternating east/west ends for the two buildings. The filter bank in the east end air intakes contained a straight filter wall, a floor-to-ceiling height of 3.7 m at the peak, with 360 total filters. The primary and secondary filters on the GDU site were installed nine months prior to the start of the study. The west air intakes contained a saw-tooth filter wall configuration with a floor-to-ceiling height of 3.7 m at the peak, with 444 total filters. Each test material was installed on the air exiting side of the evaporative cooler. Each test material was stapled to the wall above and below the evaporative cooler and all seams were fastened using wire ties. Barn 2 east air intake contained a 25.4 mm (1 in.) thick fiberglass media that was treated with an antimicrobial agent. Barn 1 west air intake contained a 3D vinyl screen. The test materials were inspected weekly to check for tears and plugged areas. If needed, the test materials were cleaned and repaired.

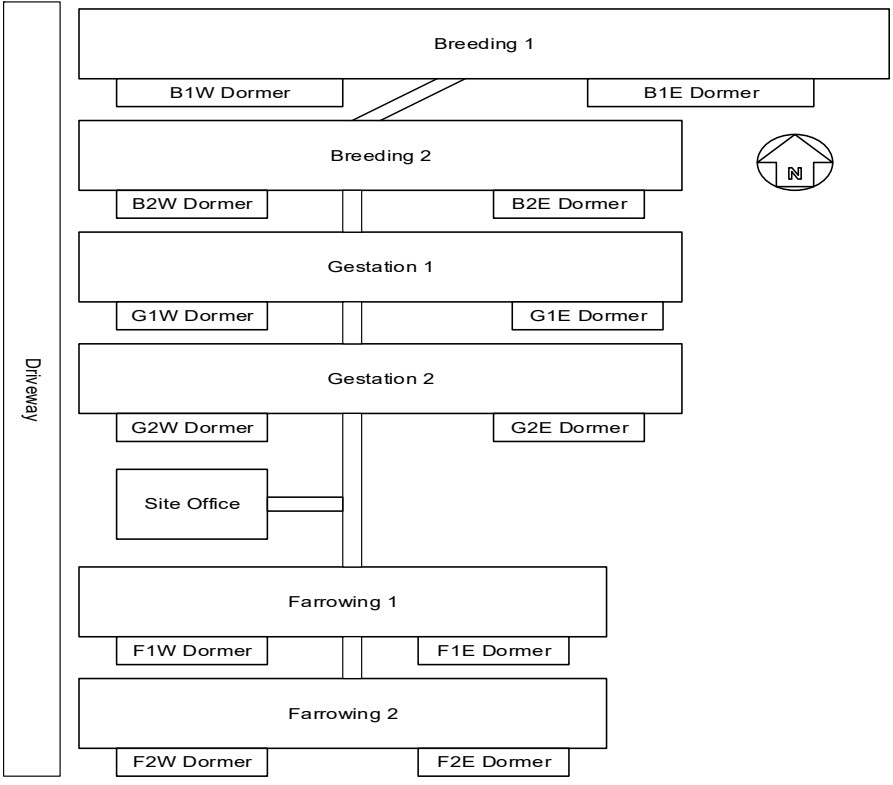

**Figure 1.** Typical layout of the breeding-gestation-farrowing sites included in the study for objectives 1 and 2. Each site had a slight variation of this layout, including driveway side and farrowing buildings on the north side. The diagram is not to scale.

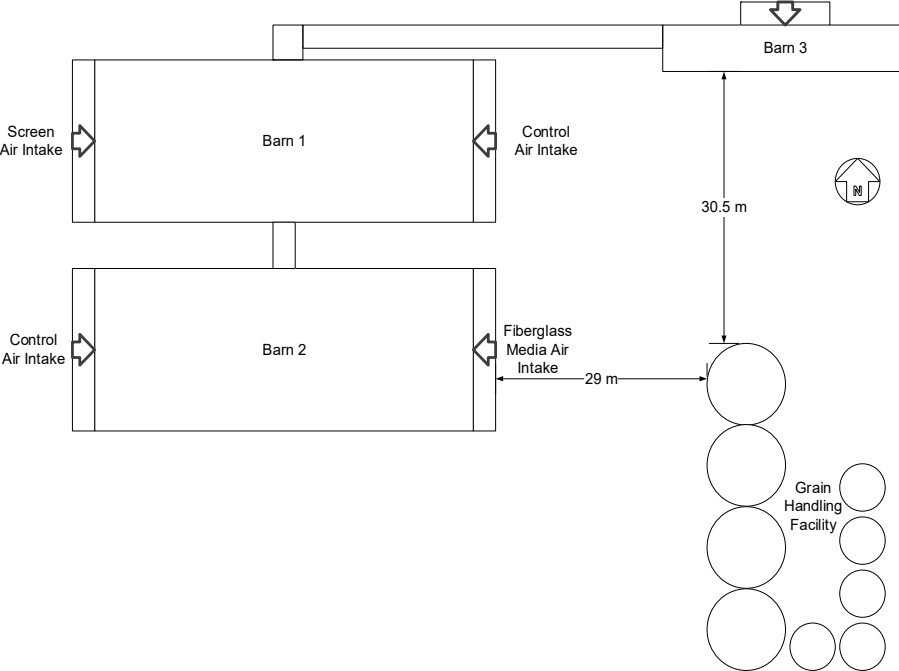

**Figure 2.** Commercial gilt development site layout utilized in the study for objective 3. Buildings one and two were utilized in the study and had identical ventilation systems. The test materials on the air intake is listed. Note drawing is not to scale. Note the grain handling facility represents nine grain bins (diagram size is proportional to actual size) with a dryer located in the center of the bins.

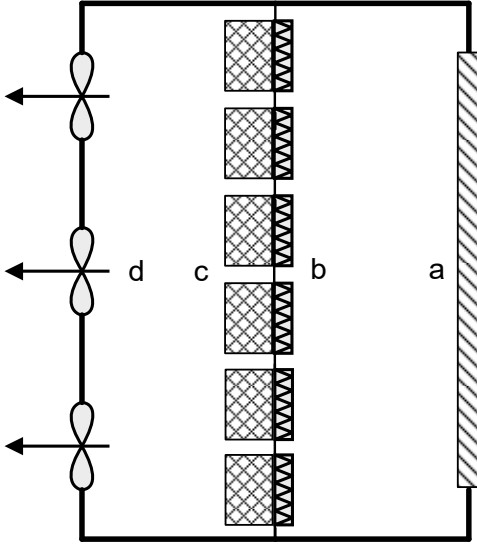

**Figure 3.** Top view of building air intakes with a straight filter wall. Air moves right to left, with the evaporative cool cell on the right (a), the primary (b) and secondary (c) filters in the center and the low pressure, high volume axial fans (d) on the left. Note the diagram is not to scale.

## 2.2. Filter Sampling and Testing

In this study, airflow capacity and the mass of primary filters were measured (i.e., secondary MERV 15 filters were not studied). Airflow was measured using the Mobile Air Filter Testing (MAFT) laboratory, with airflow measurements taken at 37 Pa pressure drop across the primary filter in series with a new v-pocket MERV 15 secondary filter (see Reference [12] for a detailed description of the technical capabilities of MAFT laboratory). At the start of testing, a reference standard pair of MERV 8 primary and MERV 15 secondary filters was tested to ensure MAFT measurements were accurate. When

testing primary filters from the site, a clean MERV 15 secondary filter was used in series. A different secondary filter was used in series dependent on the biosecurity status of the site. Sample size was based on prior field studies wherein all primary filters from one dormer were tested. The sample size was determined using a single sided t-test with a statistical power of 95%. A semivariogram of the entire dormer's filter tests determined the minimum distance between filters for independent samples. Within a dormer, filters were quasi-random sampled to meet the minimum sample size (5.5% of all filters in dormer) for the specific dormer and the minimum distance between filters.

The initial filter tests for objectives 1 and 2 were completed May–June 2017 and the end filter tests were completed September–October 2017. During both filter tests the same primary filters were sampled and re-installed in the same location within a dormer. Mass of the filters were measured prior to airflow measurements with a bench check mass scale (Model GBK 35a, Adam Equipment Company, Oxford, CT, USA) with a readability of 0.5 g. Site layout and filter factors were recorded for each site and for each individual filter tested.

Due to the lack of known spatial airflow reduction patterns for the filter wall configurations for objective 3, the minimum spacing between filters was set to one filter separation between sampled filters in the sampling plan. Primary filters were tested (14 September 2017) two days after both mitigation strategies were installed. Following testing in MAFT the primary filters were re-installed. The final mass and airflow measurements were taken on 7 November 2017 (54 days of testing).

### 2.3. Biosecurity Plan

To ensure the biosecurity of the sites utilized in this study a comprehensive biosecurity plan was developed and implemented for MAFT. The first stage in the plan was the planning of testing to move MAFT from high health status sites to low health status sites (i.e., PRRSV negative sites first then PRRSV positive sites, GDU to sow farm, etc.) for each round of tests. This practice aligned with the producer's biosecurity plan for site-to-site movements. MAFT was thoroughly cleaned and disinfected between sites that is, cleaning of the duct interior, the trailer walls, scale, power cords, all equipment used and the truck and trailer exterior. When possible, a post-cleaning downtime of 24 h was maintained between sites to reduce risk of disease transmission. The intake filters on MAFT were replaced between rounds of tests or when MAFT was taken from a PRRSV positive to a PRRSV negative farm.

### 2.4. Weather Data

Weather data was obtained from Automated Weather Observing System (AWOS) stations located near the study sites: Clarion Municipal Airport (CAV), Webster City Municipal Airport (EBS) and Iowa Falls Municipal Airport (IFA). For each site, proximity to the closest stations were considered and the weather values were averaged together depending on the site (Table 1). Average daily wind speed (km h$^{-1}$), average daily wind direction (degrees) and total liquid precipitation (cm) values were gathered for each site's specific study time frame.

**Table 1.** Breeding-gestation-farrowing sites and corresponding weather stations utilized in the study for objectives 1 and 2. The multiple weather stations for one site were average together for statistical analysis.

| Site | Station(s) | Distance, km |
|:---:|:---:|:---:|
| 1 | CAV, EBS, IFA | 21, 30, 30 |
| 2 | CAV, EBS, IFA | 25, 20, 33 |
| 3 | CAV, EBS, IFA | 11, 35, 40 |
| 4 | CAV | 8 |
| 5 | CAV | 14 |
| 6 | CAV | 15 |
| 7 | EBS | 18 |
| 8 | EBS | 10 |

*2.5. Statistical Analysis*

Statistical analyses for objectives 1 and 2, were based on results from the first and second filter tests in combination with site factors, filter factors and weather data. Filter tests were rejected if excessive water damage compromised the integrity of the primary filter construction. Airflow reduction and mass gain were calculated for each filter and then normalized to a daily airflow reduction and a daily mass gain for the study time frame. A Mixed statistical model (JMP PRO 13, SAS Institute, Cary, NC, USA) was developed to select factors that significantly impacted airflow reduction and mass gain using a backwards elimination process. Within the Mixed model the individual farm and dormer were utilized as random effects, with weather data, filter age at first test, filter factors and site factors utilized as fixed effects (Equation (1)). Statistical analysis for objective 3 utilized the reduction in airflow rates calculated for each of the 100 primary filters sampled in the study. The reduction in airflow rates were evaluated for outliers by air intake using Chauvenet's Criterion.

$$F_{ijklmnopqr} = \mu + S_i + B_j + FB_k + FO_l + DC_m + DO_n + DS_o + LP_p + WS_q + WD_r + \epsilon \qquad (1)$$

where,

| | |
|---|---|
| $F_{ijklmno}$ | observed airflow reduction rate |
| $\mu$ | grand mean airflow reduction rate |
| $S_i$ | site random effect |
| $B_j$ | building and dormer random effect |
| $FB_k$ | filter brand fixed effect |
| $FO_l$ | Filter installed orientation fixed effect |
| $DC_m$ | dormer configuration fixed effect |
| $DO_n$ | dormer orientation fixed effect |
| $DS_o$ | driveway side fixed effect |
| $LP_p$ | liquid precipitation fixed effect |
| $WS_q$ | average wind speed fixed effect |
| $WD_r$ | average wind direction fixed effect |
| $\epsilon$ | random error |

## 3. Results

*3.1. Objectives 1 and 2*

Data were initially collected on 848 filters, with 841 filters used in the final data analysis. The seven filters eliminated from the study were either damaged from water exposure or the data was not properly saved due to software issues. The interval of filter tests ranged from 113 to 140 days. Table 2 shows the five filter and site factors that were collected from the study. Moisture content of the filter media and the cardboard frame were observed to be factors impacting both the mass of the filters and the airflow. It was observed that wet filters had a lower airflow than dry. The extent of this effect was not determined due to an inability to accurately measure the moisture content of the filter media and cardboard frame.

**Table 2.** Filter and site factors evaluated in this study for objectives 1 and 2.

| **Filter Factors** | |
|---|---|
| | 1. Filter manufacturer: n = 3 |
| | 2. Filter installed orientation: correct airflow direction or backwards airflow direction |
| **Site Layout Factors** | |
| | 1. Dormer configuration: dormer faces road (r), faces small field and railroad (fr), faces field (f), faces direct exhaust outlet of adjacent barn (e), faces dormer of adjacent barn (eo), faces dormer of adjacent barn and office (do) |
| | 2. Dormer orientation: dormer faces north (n), faces south (s) |
| | 3. Relationship to driveway: dormer on driveway side (d), not on driveway side (nd) |

### 3.1.1. Mass Gain Model

Filter mass gain is considered a highly accurate predictor of filter lifespan in conventional HVAC systems, For example, one method for estimating lifespan utilizes the filter's dust holding capacity [13]. In contrast, Mixed model and correlation ($R^2$ = 0.056 for the initial tests and $R^2$ = 0.029 for the final tests) analyses found that filter mass was not a reliable indicator of filter lifespan in this application. The wide variation in the mass gain observed among filters used in swine buildings was unique when compared to more traditional filter applications but can be explained by moisture, dust and insects entrapped in the media and the cardboard frame. HVAC systems are typically designed for 100 Pa and swine buildings are designed for 37 Pa. The impact of the difference in filter operating pressure affects entrapment mechanisms is unknown at present but is a researchable question.

### 3.1.2. Airflow Reduction Model

The final Mixed model for airflow reduction showed that all filter and site layout factors were significant (Figure 4). Variables associated with weather that is, total liquid precipitation, wind speed and wind direction, were not significant but all *p*-values were less than 0.10 (Table 3). Notably, sites were proximal, thus differences in weather variables among sites was limited.

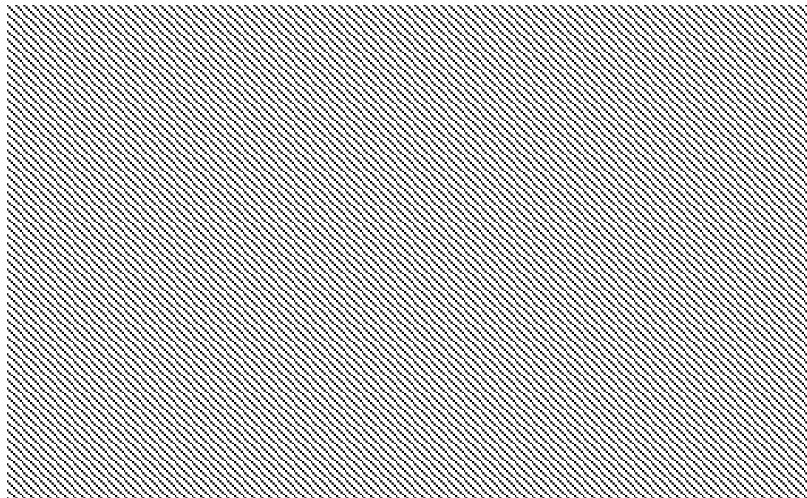

**Figure 4.** Predicted airflow versus the measured airflow values using the Mixed airflow model for objectives 1 and 2.

**Table 3.** Summary of all factors in the airflow reduction Mixed model for objectives 1 and 2. An α value of 0.05 was used. Refer to Table 2 for the different levels of the filter and site factors.

| Source | F Ratio | Prob > F |
|---|---|---|
| Filter Brand | 3.48 | 0.0314 |
| Filter Installed Orientation | 4.15 | 0.0419 |
| Dormer Configuration | 47.31 | <0.0001 |
| Dormer Orientation | 35.28 | <0.0001 |
| Driveway Side | 10.93 | 0.0010 |
| Liquid precip (cm) | 5.39 | 0.0803 |
| Av. Wind Speed (kmh) | 6.39 | 0.0649 |
| Average wind direction (deg) | 5.38 | 0.0785 |

Airflow reduction rates varied among buildings and dormers across all eight sites (Figure 5). The building and dormer average variation was attributed to the variation in site layouts for driveway side and dormer configuration (facing exhaust outlets, fields, roads). The next sections will discuss each observed variable's impact in-depth.

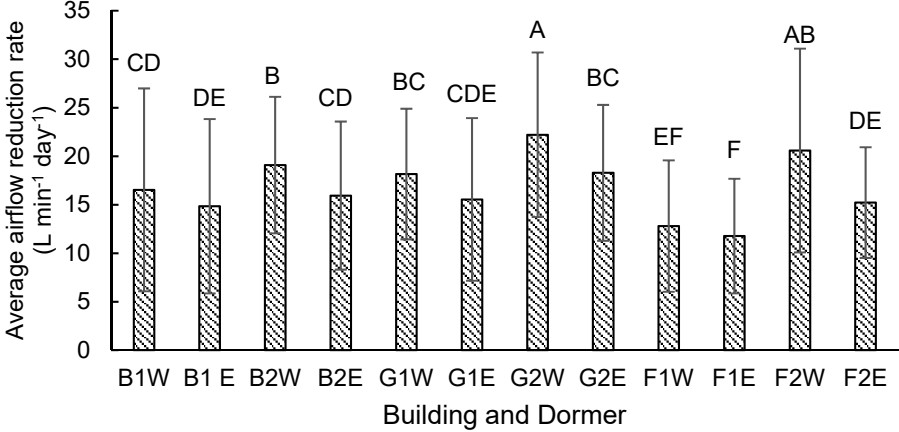

**Figure 5.** Barn and dormer average airflow reduction from the breeding-gestation-farrowing sites for objectives 1 and 2. Error bars represent one standard deviation. Bars not connected by the same letter are significantly different $\alpha = 0.05$. Refer to Figure 1 for the building and dormer location represented as the *x*-axis codes. The barns are: Breeding 1, Breeding 2, Gestation 1, Gestation 2, Farrowing 1 and Farrowing 2; each barn has an east and west dormer.

### 3.1.3. Filter Brand and Installed Orientation

Among the three filter brands evaluated in the study, a significant difference was observed between Supplier A and Supplier B ($p = 0.0086$). No statistical difference was observed between Supplier C and Suppliers A or B. The installed orientation of the filter (manufacturer intended airflow or reverse airflow direction) also showed a significant difference, with backwards installation having a higher airflow reduction rate ($p = 0.0419$). The significance of the filter brand and the installed orientation on the airflow reduction rate support the need for attention to detail in both the purchasing and installation of filters. While all three brands were marketed as MERV 8 high capacity primary filters, the different airflow reduction rates suggested subtle differences in the quality of the media.

### 3.1.4. Dormer Configuration

Among the dormer configurations noted in the study, filters facing the exhaust outlets of adjacent buildings experienced a significantly higher airflow reduction rate compared to all other configurations, $p = 0.0105$, (Figure 6).

Mold growth on the filter media face, likely due to the warm moist air exiting the adjacent building, was primarily observed on filters in the exhaust dormers and contributed to a reduction in airflow. Dormers not facing exhaust outlets exhibited little mold growth. The only other configuration associated reduced airflow was a dormer facing a small field and railroad; but this configuration was only observed on one building.

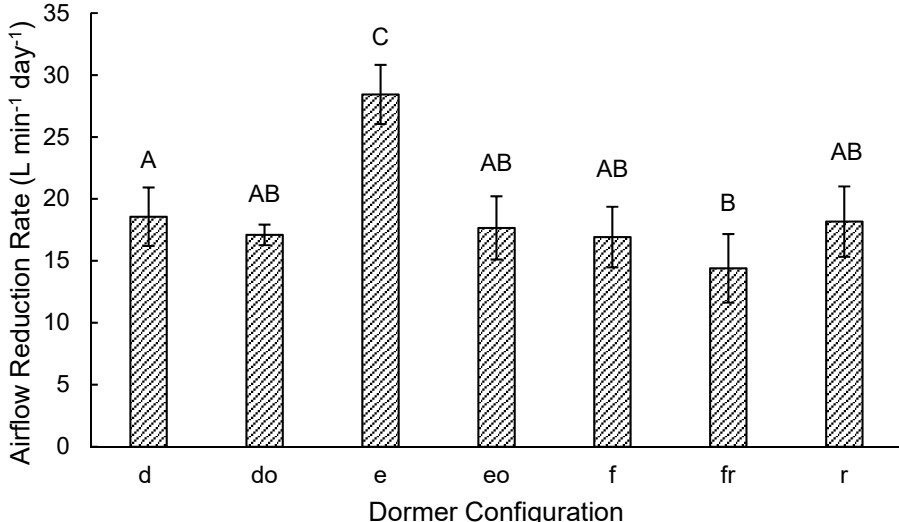

**Figure 6.** Airflow reduction rate by dormer configuration from the breeding-gestation-farrowing sites for objectives 1 and 2, error bars represent one standard error. Bars not connected by the same letter are significantly different, $\alpha = 0.05$. The dormer configurations are as follows: r = faces road, fr = faces small field and railroad, f = faces field, e = faces direct exhaust outlet of adjacent barn, d = faces dormer of adjacent barn, eo = faces exhaust outlet of adjacent barn and office, do = faces dormer of adjacent barn and office.

### 3.1.5. Dormer Orientation

An analysis of dormer orientation showed that north-facing dormers had a significantly higher airflow reduction rate than south-facing dormers ($p = 0.001$), despite the fact that the average wind direction over the study timeframe was out of the south. One possible explanation is that the southerly wind produced a vacuum and induced swirling and entraining of inward dust on the north side of the building. Depending on the site layout, the location of windbreaks may have also influenced this finding.

### 3.1.6. Driveway Side

Filters in a dormer near the driveway had a significantly higher airflow reduction rate than did those away from the drive ($p = 0.001$). It is not surprising that filters nearer the driveway had a high airflow reduction rate, as gravel roads are a major source of dust.

### *3.2. Objective 3*

In this study the primary filters were on test for 54 days during the Fall row crop harvest season. Over the course of the study period neither test material experienced blockage that warranted cleaning or replacement, though the fiberglass media did require minor spot repairs during the study. Under the conditions experienced during this period, both test materials significantly reduced the airflow reduction rate compared to the controls ($p < 0.0001$; Figure 7), although no difference was detected between test materials ($p = 0.095$). Utilizing an airflow reduction cutoff of 4248 L min$^{-1}$, the end-of-life estimate for the primary filters evaluated in this study are shown in Table 4. The end-of-life estimates are very different between the fiberglass media and the 3D vinyl screen, though an economic value was not determined.

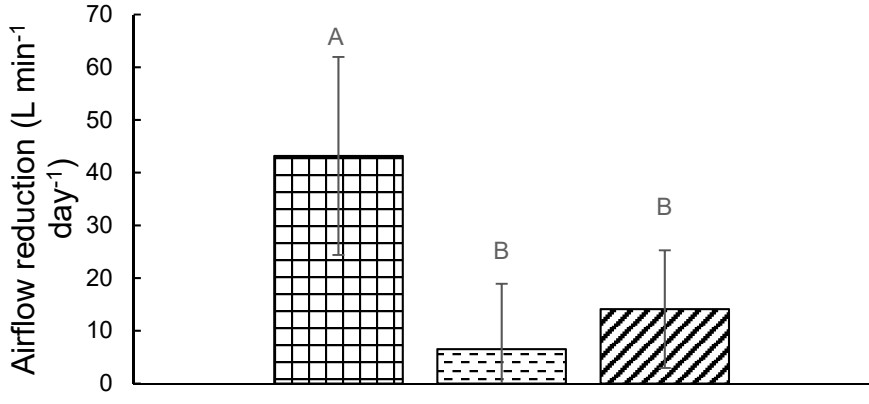

Figure 7. Airflow reduction rate from the gilt development unit (GDU) site for objective 3. The error bars represent one standard deviation. Bars not connected by the same letter are significantly different, $\alpha = 0.05$.

**Table 4.** Average airflow reduction rate from the GDU site for objective 3 from each experimental group with the end-of-life estimate with an airflow reduction cutoff of 4248 L min$^{-1}$ from new.

| Experimental Group | Average Airflow Reduction Rate (L min$^{-1}$) ± SD | End-of-Life Estimate (Days) |
| --- | --- | --- |
| Control | 43.16 ± 18.78 | 98 |
| Fiberglass media | 6.51 ± 12.40 | 653 |
| 3D vinyl Screen | 14.10 ± 11.17 | 301 |

## 4. Practical Implications

This study identified key factors that impacted the filter airflow reduction rate on a swine building on a per day basis. An estimate of total airflow reduction (Table 5) also allows for the estimation of airflow reduction per animal. For example, an 800 head gestation barn with 312 installed filters and an estimated maximum ventilation rate of 9167 L min$^{-1}$ per sow experienced an airflow reduction of 7.8% to 15.3%. For the purpose of estimating end-of-life, a maximum allowable airflow reduction per filter must be determined or assumed. This value would depend on the building's specific design and the producer's desire to avoid heat stress in the building. For the calculations presented herein, a maximum airflow reduction of 4248 L min$^{-1}$ from new was used to estimate the useful life of the primary filters. Note that this estimate is for a new pair of primary and secondary filters and would need to be adjusted to account for the secondary filters loaded simultaneously for future change outs. Table 6 shows the mean airflow reduction rate with the highest and lowest configuration estimates along with the experimental groups from objective 3 and the respective useful life estimates. A key assumption in this estimate is that the dust loading potential is constant, regardless of time of year. Without annual data on airflow reduction, the estimate will be difficult to predict. Another key assumption with this is that the ventilation rate is constant, maximum ventilation rate, which annually will vary with ambient conditions. These estimates also assume that the cardboard frame of the primary filter will hold up to the environmental conditions for that timeframe. With further data, a more accurate model can be developed to determine filter end-of-life.

**Table 5.** Total airflow reduction for all factors from the breeding-gestation-farrowing sites for objectives 1 and 2 observed in the study and the predicted airflow reduction per sow for an 800 head gestation barn with 312 filters (similar to the gestation barns in the study).

| Factor | Total Airflow Reduction per Filter (L min$^{-1}$) | Airflow Reduction per Animal (L min$^{-1}$ sow$^{-1}$) (800 Head, 312 Filters) |
|---|---|---|
| Supplier A | 2219 | 865.9 |
| Supplier B | 2595 | 1011.7 |
| Supplier C | 2276 | 886.9 |
| Correct filter orientation | 2041 | 795.8 |
| Incorrect filter orientation | 2686 | 1047.5 |
| Road configuration | 2276 | 887.6 |
| Field and railroad configuration | 1824 | 711.4 |
| Field configuration | 2139 | 834.2 |
| Exhaust configuration | 3590 | 1400.0 |
| Dormer configuration | 2356 | 918.8 |
| Exhaust and office configuration | 2202 | 585.8 |
| Dormer and office configuration | 2156 | 840.8 |
| North dormer orientation | 2579 | 1005.8 |
| South dormer orientation | 2147 | 888.3 |
| Driveway side | 2449 | 955.0 |
| Non driveway side | 2278 | 888.3 |

**Table 6.** Primary filter end-of-life estimates based on the study findings using an airflow reduction of 4248 L min$^{-1}$ as the end of life cutoff for objectives 1, 2 and 3 from this study.

| Setup/Experimental Group | Airflow Reduction Rate (L min$^{-1}$ day$^{-1}$) ± 95% CI | End-of-Life Estimate (Days) |
|---|---|---|
| Study average (objectives 1 and 2) | 16.67 ± 0.57 | 255 |
| Exhaust configuration | 28.43 ± 4.66 | 149 |
| Field and railroad configuration | 14.39 ± 5.41 | 295 |
| Control (objective 3) | 43.16 ± 18.78 | 98 |
| Fiberglass media (objective 3) | 6.51 ± 12.40 | 653 |
| 3D vinyl Screen (objective 3) | 14.10 ± 11.17 | 301 |

## 5. Conclusions

Field measurement of the mass gain and airflow reduction of primary filters in swine building ventilation systems was completed for one specific type of ventilation system on a sample population of nearly identically-constructed sites. A wide variation in filter mass gain was seen, with the accumulation of moisture, dust and insects in the media and cardboard frame indicated as factors that impacted the filter mass gain. All filter and site layout factors recorded in the study were found to have a significant effect on the airflow reduction rate, whereas weather data, (total liquid precipitation, wind speed and wind direction) were not important variables, $p = 0.0803$, $0.0649$, $0.0785$, respectively. Filter brand and correct orientation at the time of installation were important areas of consideration, $p = 0.0314$ and $0.0419$, respectively. Site-specific layout factors highlight the consideration of air intake location on buildings relative to dust sources and their impact on airflow reduction rate per day over the summer months. Significant factors included dormer configuration ($p < 0.0001$), dormer orientation ($p < 0.0001$) and driveway side ($p = 0.001$). Further research examining the airflow reduction rate throughout an entire year is needed to evaluate filter lifespan for both primary and secondary filters. Research involving the different ventilation systems with filtration should be studied as the systems vary widely in design and operation.

For the high dust-loading site, no significant difference was found between the two test materials, vinyl screen and fiberglass media, for either airflow reduction rate but both significantly reduced the airflow reduction rate compared to the control group. Further evaluation of test materials is warranted

to better quantify the effects of each test material on high dust loading sites. An economic analysis should also be completed to quantify the cost reduction potential for each material. This analysis will likely be the driving force in a producer's choice on sites with high dust loading potential.

**Author Contributions:** For this work on the conceptualization, and methodology B.S., S.H., J.H., D.A., and J.S.; data collection and data preprocessing, B.S.; formal analysis, B.S., D.A., and J.Z.; writing—original draft preparation, B.S., S.H., and J.Z.; writing—review and editing, B.S., S.H., J.H, D.A., J.Z., and J.S.; supervision, S.H.; project administration, S.H, J.H.; funding acquisition, J.S.

**Funding:** This research was funded through in-kind donations from Iowa Select Farms.

**Acknowledgments:** The authors would like to thank Wyatt Murphy and Blake Fonken for the assistance during the data collection for this work.

**Conflicts of Interest:** The authors declare no conflict of interest.

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
