# Peer review of "Quantification of Site Layout and Filter Characteristics on Primary Filter Airflow Reduction on Commercial Swine Sites in Iowa"

_agriengineering, doi:10.3390/agriengineering1020022_

Round 1
Reviewer 1 Report
Overall the article provided usuable insights on the effects of dust associated with the primary (pre-filters) filters on overall ventilation performance. Replacement of primary filters have been a ongoing concern for producers as to when these should be replaced. A common rule, at the very least, is to replace all primary filters in the spring before hot weather ventilation is required. The research shows significant reduction in ventilation performance. However, given an average design ventilation rate of 9,167 L min-1 per sow, this average reductions may not be noticeable in the overall ventilation performance of the operation. The 4,428 L min-1 is a reasonable number to indicate primary (pre-filter) changeout and would significantly affect ventilation performance if these filters would be left in place.
Author Response
Point 1: Overall the article provided usable insights on the effects of dust associated with the primary (pre-filters) filters on overall ventilation performance. Replacement of primary filters have been a ongoing concern for producers as to when these should be replaced. A common rule, at the very least, is to replace all primary filters in the spring before hot weather ventilation is required. The research shows significant reduction in ventilation performance. However, given an average design ventilation rate of 9,167 L min-1 per sow, this average reductions may not be noticeable in the overall ventilation performance of the operation. The 4,428 L min-1 is a reasonable number to indicate primary (pre-filter) change out and would significantly affect ventilation performance if these filters would be left in place.
Response 1: Thank you for your time and effort in reviewing this manuscript. We made no changes based on these comments.

Reviewer 2 Report
The manuscript provides useful information regarding an important bio-security aspect at times forgotten such as airflow rate inside pig barns. I commend the authors for their work and I only have some minor comments mainly regarding the captions of figures and tables. At the moment they do not stand alone nor do they properly describe the message they want to convey. Abbreviations appearing in Tables/figures sometimes need to be described as a footnote
-Figure 2 is showing pretty much the same information as figure 1 and thus i could be removed
-It was not clear to me if all filters were newly installed for this study/what was the age of the filters?
-Table 2 is very confusing, I understand the authors are just listing the site and filter factors recorded but at first glance it seems like, for instance, for supplier A, filter orientation was F-correct install and so on. The table would benefit if re-organise in amore clear way
-I get the authors cannot name the filter brands but some more details about each brand are needed to clearly understand the different results for different brands
-the equation shown in L195 should be moved to the section where statistical methods are described
-figure 8: it is not possible to differentiate between treatments, could you use a different pattern/fill for each of them?
Author Response
Point 1: The manuscript provides useful information regarding an important bio-security aspect at times forgotten such as airflow rate inside pig barns. I commend the authors for their work and I only have some minor comments mainly regarding the captions of figures and tables. At the moment they do not stand alone nor do they properly describe the message they want to convey. Abbreviations appearing in Tables/figures sometimes need to be described as a footnote

Response 1: Thank you for your time and effort in reviewing this manuscript. We have updated the figure and table captions to add detail and clarify. Following the AgriEngineering template we did not add footnotes as recommended to figures 5 and 6.
Point 2: Figure 2 is showing pretty much the same information as figure 1 and thus i could be removed
Response 2: We agree, Figure 2 was removed from the manuscript and all figure numbers were updated accordingly.
Point 3: It was not clear to me if all filters were newly installed for this study/what was the age of the filters?
Response 3:. The time in operation on site was clarified to state when filters were installed relative to the start of the study. See lines 87 through 91 and line 96
Point 4: Table 2 is very confusing, I understand the authors are just listing the site and filter factors recorded but at first glance it seems like, for instance, for supplier A, filter orientation was F-correct install and so on. The table would benefit if re-organise in a more clear way
Response 4: Table 2 has been rearranged to clarify the list of filter and site factors evaluated in the study. See line 194 for the update Table 2.
Point 5: I get the authors cannot name the filter brands but some more details about each brand are needed to clearly understand the different results for different brands
Response 5: An additional sentence has been added to the site description section to clarify the differences between the brands reported in the study. The filter brands evaluated in the study were the same specifications, but the quality of the brands varied widely.
Point 6: the equation shown in L195 should be moved to the section where statistical methods are described
Response 6: Agreed. The equation was moved from the results section to the methods section in the statistical analysis subsection.
Point 7: figure 8: it is not possible to differentiate between treatments, could you use a different pattern/fill for each of them?
Response 7: Agreed. The fill for the bars were changed to easily distinguish between treatments.
